# Influence of Flight Height and Image Sensor on the Quality of the UAS Orthophotos for Cadastral Survey Purposes

Hrvoje Sertić [1], Rinaldo Paar [2,*], Hrvoje Tomić [2] and Fabijan Ravlić [3]

1   Geo-Land d.o.o., Antuna Mihanovića 14, 10450 Jastrebarsko, Croatia
2   Department of Applied Geodesy, Faculty of Geodesy, University of Zagreb, Kačićeva 26, 10000 Zagreb, Croatia
3   Dibit Messtechnik GmbH, Framsweg 16, 6020 Innsbruck, Austria
*   Correspondence: rinaldo.paar@geof.unizg.hr

**Abstract:** The possibility of using unmanned aircraft systems (UAS) for cadastral survey purposes was investigated in this research. A study site consisting of 26 ground control points (GCP) and checkpoints (CP) was established. The study site was first measured by the classical methods of geodetic surveying, i.e., by the polar method using a total station. After that, all points were additionally measured by the Global Navigation Satellite System (GNSS) Real-Time Kinematic (RTK) method. The GNSS RTK method was used to determine the coordinates of all points in the official map projection of Croatia, HTRS96/TM, while the polar method was used to increase the positional "strength" of points in all directions, i.e., to improve the relative accuracy between them. Using UASs with different image sensor characteristics, the study site was measured by an aerial photogrammetry method at different flight heights with the purpose of obtaining a high-quality digital orthophoto plan (DOF). The absolute orientation of the model was performed using the external orientation data of each digital image based on GNSS and Inertial Measurement Unit (IMU) UAS's sensors, as well as using GCPs. Achieved precision of obtained DOF, as well as accuracy analysis of aerial photogrammetry was performed by considering the adjusted survey data collected by classical and GNSS RTK methods as true values and comparing them with the coordinates obtained by the aerial photogrammetry method from DOFs. Based on the achieved results and conclusions obtained from the study site, the second field test was performed above a small settlement which served as an area for cadastral survey using the UAS and GNSS RTK method. Again, precision and accuracy were determined, based on which we derived recommendations and conclusions for using UASs for cadastral survey purposes.

**Keywords:** unmanned aircraft system; digital orthophoto; photogrammetry; accuracy; precision; cadastral survey; polar method; GNSS RTK



## 1. Introduction

The impact of the development of technology, especially in the last thirty years, on the surveying profession is enormous. It has greatly changed the work speed and approach for surveyors and enabled much greater flexibility and efficiency while solving new tasks that were challenging in the years before. In the 1990s, we witnessed the third geodetic revolution in the form of the development of Global Positioning System (GPS) technology and the application of computers and user programs with different tools. Parallel to the development of GPS, classical geodetic instruments—total stations with a series of integrated sensors—have also been developed, making it possible to solve different cadastral and engineering tasks faster and more easily than before. With this technological development of geodetic instruments due to the rapid hardware development, these different sensor classes, each with their specific advantages, can be unified, utilized, and fused as one single (nearly) universal geodetic instrument [1]. In the last ten years, there has been a rapid development of UASs, computers and data processing programs that

have become available and affordable even to small-sized surveying companies, which was not the case before. Some would say that we are witnessing the fourth geodetic revolution. According to [2], unmanned aircraft (UA) means any aircraft operating or designed to operate autonomously or to be piloted remotely without a pilot on board, while UAS means an unmanned aircraft and the equipment to control it remotely. Today, UASs have wide applications in various branches such as forestry, agronomy, geodesy, civil protection and many others. They have surpassed their original military purpose. The main advantage of using a UAS for geodetic surveying is the ability to collect a large amount of data in a short period compared to classical surveying methods. Although UASs are still rarely used in cadastral and geodetic surveying of terrain in Croatia to obtain geodetic site plans that are a representation of the actual state of the terrain in terms of position and height, the growing trend of using UASs for this purpose is visibly increasing.

As in Croatia, the growing trend of using UASs for cadastral survey applications is also visible in other countries. UASs and their use in the surveying profession have been a subject in many publications and studies. Accuracy of UAS photogrammetry and the "Structure-from-Motion" (SfM) method using a photogrammetry survey as a function of the number and location of GCPs used were investigated by [3]. The case study involved an area of 1200+ ha with 100+ GCP and 2500+ photos. The result of the study demonstrates that UAS-SfM photogrammetry accuracy depends on the location and number of GCPs introduced in the bundle adjustment (BA). With a higher number of GCPs (more than 2 GCPs per 100 photos in that case study), the RMSE have converged slowly to a value approximately twice the average Ground Sampling Distance (GSD). It is also stated that GCPs should be evenly distributed around the whole interest area, ideally in a triangular mesh grid, since, with this setup, the maximum distance to any GCP is minimized. Different distributions of GCPs have been studied by [4] in an attempt to optimize the products obtained by UAS photogrammetry. The case study involved an area of 17.64 ha. In that study, the best results were obtained with edge distribution and stratified distribution. A case study involving an urban area of about 1 ha was mapped in a study [5] to determine the suitable number of GCPs for UAS images georeferencing by varying number and spatial distribution. The study, among other things, concludes that GCPs in the corners are essential; a stratified random placement of control points offers a similar accuracy and an even better one than a systematic placement. An increase in the number of control points leads to improved accuracy as the accuracy converges to the value of two GSDs in planimetry and three GSDs in elevation. Three image acquisition flights were performed for two sites of a different character (urban and rural) along with three calculation variants for each flight: georeferencing using only ground-measured GCPs, GNSS RTK only, and a combination was performed in the study by [6]. The best model resulted from a combination of both GCPs and GNSS RTK UAS coordinates for the calculations.

The study in [7], using imagery from six study areas across Europe and Africa, suggested that scene context, flight configuration, and GCP setup significantly impact the final data quality and subsequent automatic delineation of visual cadastral boundaries. A large image overlap, as well as a cross-flight pattern, increases the accuracy and completeness of automatically delineated walls. The possibility of using UAS photogrammetry and laser scanning for cadastral mapping in the Czech Republic was discussed in the paper by [8] using data from 12 test areas. The required accuracy of 0.14 m was achieved for more than 80% of points in the case of the image point clouds and orthoimages, and more than 98% in the case of the LiDAR point cloud. The use of UAS for updating farmland cadastral data in areas subject to landslides was studied by [9]. The study site covered an area of more than 70 ha at the site of a landslide and the accuracy of a derived digital orthophoto map and Digital Surface Model (DSM) was analysed. The coordinates of the GCPs were determined using two independent methods, static GNSS and RTK GNSS. Slightly better results were obtained when using static GNSS but given the amount of time and labour required for conducting such a survey, this method cannot be considered to

produce significantly better results than the RTK GNSS method. Although the achieved accuracy of the digital orthophoto map met the required 0.10 m according to Polish laws and regulations using 8 GCPs, it is advised that a greater number of GCPs are used to increase the certainty that the photogrammetric products generated are correct and as accurate. The use of UASs in cadastral surveying was investigated by [10], where UASs were tested for capturing geodata and compared with conventional data acquisition methods. Two study sites were used and surveyed with a tachymeter-GNSS combination as well as a UAS and the results were compared with the requirements of the Swiss cadastre. In Switzerland, when discussing cadastral surveying, there are five zones with different levels of surveying tolerances. Regarding those zones, the first study site represents "Intensively used agricultural and forested areas", while the second study site represents "Built-up areas and construction zones". The average accuracy of 2.3 cm (horizontal) and 3.8 cm (vertical) was achieved, and the Swiss cadastre requirements were met regarding the measurement of ground covers and individual objects such as buildings, roads, paths, sidewalks, fields, gardens, water, or forest edges. How the use of UAS can speed up the process of land registration in developing countries such as Indonesia, where the amount of registered land is low, was shown and detailly described by [11]. The study area was chosen based on the following criteria: low amount of registered land, non-conflict area and not customary land. The chosen study area stretches across 4706 Ha with only 12.22% of land registered. The approach, which included the high involvement of the community in terms of delineation of parcel boundaries in the field from the derived true orthophoto map, proved to be more efficient in generating more parcels in an adjacent area.

The literature review revealed various flight configurations and GCP setups to meet accuracy requirements as regulated by national laws or other legislation concerning the use of SfM to register authoritative geospatial data. The geospatial data obtained in this manner may support the implementation of FIGs (International Federation of Surveyors), a so-called "Fit-for-purpose" approach to land administration systems (national authoritative geospatial datasets), meaning that each land administration system must be designed to manage current land issues rather than simply follow advanced technical standards [12]. The aim of this research is to determine whether the UAS orthophotos can be used for cadastral survey purposes in Croatia and to determine how the methodology in the terms of using different flight heights and image sensors reflects on the achieved accuracy and precision.

## 2. Materials and Methods

Wide availability, simplicity and increasing applicability of measurements with various integrations of active (GNSS RTK, IMU, LiDAR) and passive image sensors enable the relatively fast and efficient collection of a large amount of accurate and reliable spatial information. These sensors can be integrated into a classical surveying instrument or installed on a drone or other vehicles. The development of computer technologies for some time enables almost completely automatic processing of some of this collected spatial information using the photogrammetric SfM method. With this method, it is possible to create different raster and vector spatial models: digital orthophoto maps, point clouds, mesh, raster and vector terrain models such as digital terrain model (DTM) and digital elevation model (DEM), classified raster and vector data and different indices determined by combinations of image features in different spectrum areas. These different raster and vector spatial models are usually used by geodetic experts to make spatial plans for design and other purposes [13–15].

A geodetic survey using UAS, and classical methods of geodetic surveying was carried out on two independent research sites. The first study site was set up in the city of Samobor, Croatia, with the field being a flat asphalt surface. The second study site was set up above a small settlement in the city of Jastrebarsko, Croatia, where acknowledgements regarding obtained accuracy and precision from the first field test were applied.

The field of UASs in Croatia is governed by Commission Delegated Regulation (EU) [2] and Commission Implementing Regulation (EU) [16]. These regulations directly replaced national regulations, so the Ordinance on unmanned aircraft systems [17] has been repealed. Commission Delegated Regulation (EU) [2] addresses technical requirements, operator requirements and requirements for third-country operators, while Commission Implementing Regulation (EU) [16] addresses operational requirements and registration. Before the test flights, the UAS operator completed an online course on the website of the Croatian Civil Aviation Agency [18] and obtained a certificate of competency and registered. Based on the request of the Faculty of Geodesy, University of Zagreb, the State Geodetic Administration issued a Decision approving aerial photogrammetry of a part of the public area and cadastral parcel no. 2690/16 in the cadastral municipality Samobor which was used as a research site for this study (Figure 1). The decision was issued in accordance with Article 98, paragraph 3 of the Defence Act [19] and Article 9 of the Regulation on Aerial Photography [20]. The city of Samobor as the owner of the cadastral parcel issued consent for aerial photogrammetry based on Article 6, paragraph 1 of the Regulation on Aerial Photography [20]. Flight operations were performed in the "open" subcategory A2 within the field of view of the remote pilot, with a maximum flight height of 120 m. The registration of the flight operations, i.e., the establishment of the ad hoc structure, was performed using the mobile application of the Airspace Management Cell (AMC) portal in accordance with the Ordinance on Airspace Management [21].

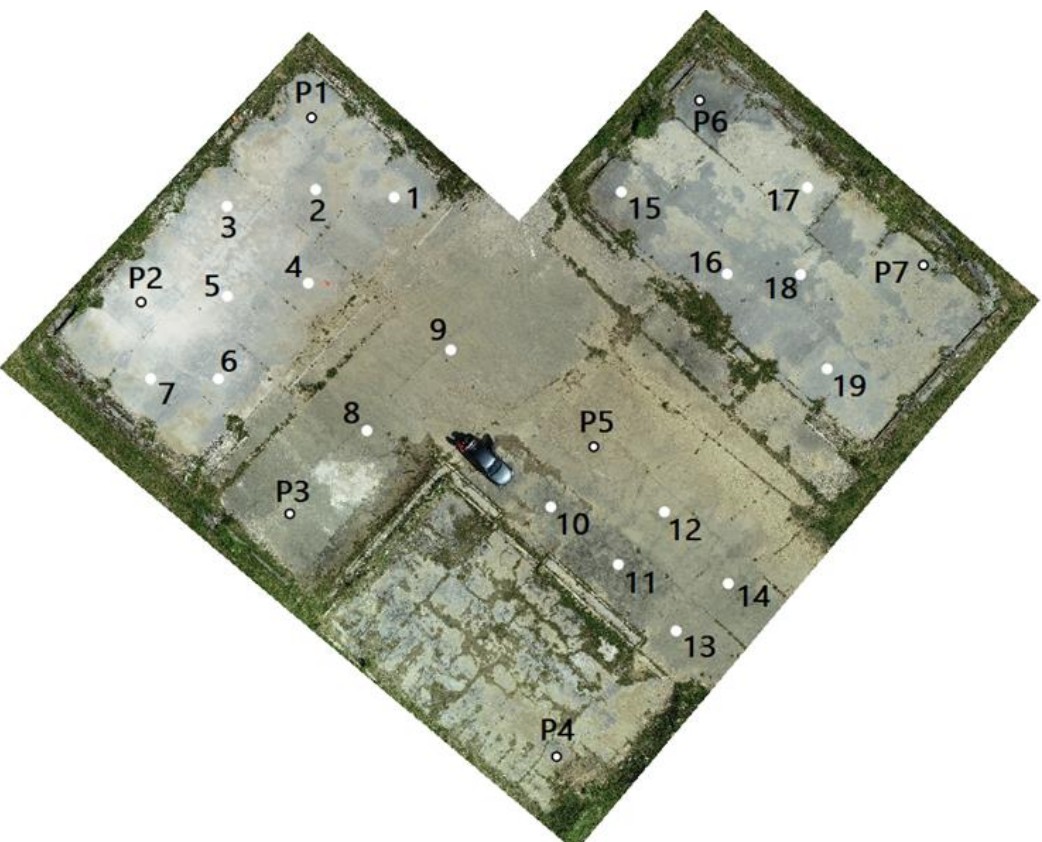

**Figure 1.** Established research site in the city of Samobor, Croatia, with marked GCPs (P1–P7) and CPs (1–19).

To investigate the possibility of using unmanned aircraft for the purpose of geodetic surveying, a research site in the city of Samobor was established. The research site consisted of 7 GCPs and 19 CPs, in the part of cadastral parcel no. 2690/16 in the cadastral municipality Samobor, as shown in Figure 1. GCPs and CPs were drawn using a previously made point model and painted with white spray on the asphalt surface as circles with a diameter

of 20 cm, and a black border was added to the GCPs as shown in Figure 2. The study site was first measured by the classical method of geodetic surveying, the polar method, using a total station. Leica TCRP1201 robotic total station was used, which has a standard deviation of the accuracy of determining angles of 1" according to ISO standard 17123-3 and the accuracy of determining distances of 2 mm + 2 ppm according to ISO standard 17123-4 [22]. After that, all points, GCPs and CPs, were measured by the GNSS RTK method using a Topcon HiPer SR receiver, which in RTK measurement mode has a horizontal accuracy of 10 mm + 1.0 ppm, and a vertical accuracy of 15 mm + 1.0 ppm [23]. GCPs were measured in two independent repetitions, where each repetition had three consecutive measurements, and each measurement epoch had a duration of 30 s. In accordance with the Ordinance on the method of performing basic geodetic work [24], all GCPs were measured again in the same way after two hours. CPs were measured once in a way that each measurement had a duration of six seconds. The GNSS RTK method was used to determine the coordinates of all points in the official map projection HTRS96/TM, and the polar method to increase the positional "strength" of points in all directions, i.e., to increase the relative accuracy between them. By adjusting all the measurements from both, the polar method and GNSS RTK survey, the coordinates of the measured points collected by classical methods were obtained. Those coordinates were taken as error-free in the a posteriori performed analysis due to relatively low theoretical deviations from the theoretical deviations obtained by the aerial photogrammetry method. CPs were used to assess the absolute accuracy of the models. With CPs, it is possible to obtain the difference between the reconstructed model and the actual position of points.

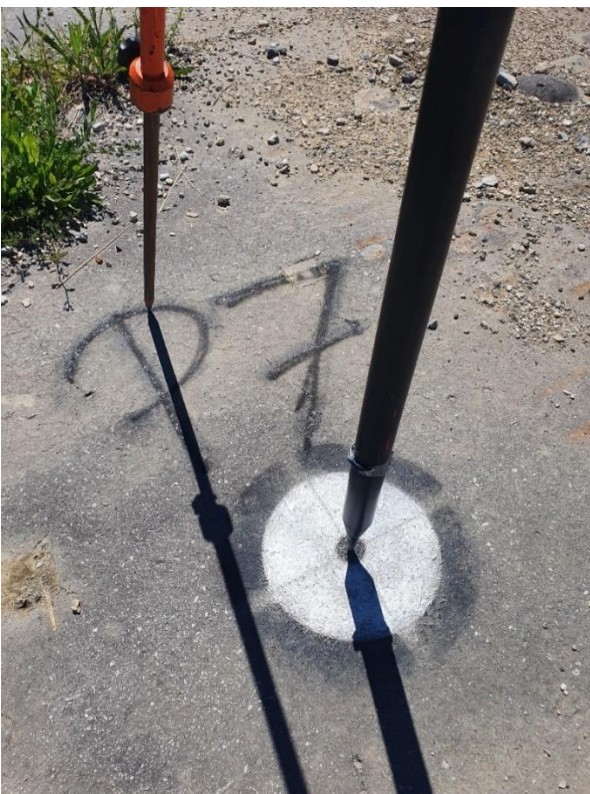

**Figure 2.** Marked GCP on the study site in the city of Samobor, Croatia.

To obtain the highest possible statistical accuracy of the GCPs and CPs, the point coordinates were adjusted by the least-squares method, the Gauss–Markov linear model. The Gauss–Markov adjustment model is based on the idea that the unknown parameters (in our study only horizontal coordinates E, N) are estimated with maximum probability. Three independent network adjustments were carried out in the JAG3D (Java Applied

Geodesy 3D) software package. Terrestrial observations: distances and angles collected using the polar method and datum points; coordinates measured by a GNSS RTK method in a micro-network estimated the statistical best-fit solution for the coordinates. The estimated standard deviation of GCPs and CPs coordinates is in a range of a few millimetres. If the accuracy of the reference point coordinates is, for example, better by a factor of 5–10 than the photogrammetric point determination and there are no inconsistencies in reference point coordinates, then reference point coordinates can be introduced as error-free. Logically, this approach leads to an error-free definition of the datum, with bundle adjustment input being zero standard deviation measurements [25].

A UAS photogrammetry survey of the study site was performed using four different UASs with different characteristics of the image sensor and with or without the GNSS RTK sensor. Flights were taken at two different flying heights above ground level: at 60 m and 120 m, with front and side overlaps of 80% and a camera angle of 90° (vertical aerial images). The collected images were processed in 3Dsurvey software in ten different projects. Bundle adjustment was carried out using Global mode in the 3Dsurvey software. The software offers two more bundle adjustment modes: incremental and hybrid (global + incremental mode). As for the dense reconstruction, the software offers four reconstruction levels (low, medium, high, and extreme), which relates to the detailedness of the reconstructed point cloud. In this study, a high level of reconstruction was used.

Those ten projects (Table 1) differ in the used UAS and their image sensor characteristics and GNSS RTK sensor (Table 2). Furthermore, they differ by flight heights and UAS flight speeds, and by the absolute orientation of the model (Table 1). UASs DJI Phantom 4 and DJI Phantom 4 Pro V2.0 does not have an integrated GNSS RTK sensor, so the absolute orientation of the models was conducted by georeferencing the model with GCPs for two flight heights at 60 m and 120 m (four projects). UASs DJI Phantom 4 RTK and senseFly eBee Plus have integrated GNSS RTK sensors and the absolute orientation of the models were carried out with and without the use of GCPs for two flight heights at 60 m and 120 m (six projects). Since the UASs DJI Phantom 4 Pro V2.0 and DJI Phantom 4 RTK have the same image sensor characteristics, the absolute orientation with GCPs was not carried out for the DJI Phantom 4 RTK, just for DJI Phantom 4 Pro V2.0. Altogether, ten different projects have been made (Table 1). The result of the processing of each project is a 3D point cloud from which, after classification, a DTM was made, and finally a DOF map.

**Table 1.** Overview of the basic project parameters.

| | Project 1 | Project 2 | Project 3 | Project 4 | Project 5 | Project 6 | Project 7 | Project 8 | Project 9 | Project 10 |
|---|---|---|---|---|---|---|---|---|---|---|
| **UAS** | Phantom 4 | Phantom 4 | Phantom 4 Pro V2.0 | Phantom 4 Pro V2.0 | Phantom 4 RTK | Phantom 4 RTK | eBee Plus | eBee Plus | eBee Plus | eBee Plus |
| **GCP** | + | + | + | + | − | − | + | + | − | − |
| **RTK** | − | − | − | − | + | + | − | − | + | + |
| **60 m** | + | − | + | − | + | − | + | − | + | − |
| **120 m** | − | + | − | + | − | + | − | + | − | + |
| **UAS Speed** | 5 m/s | 5 m/s | 6 m/s | 6 m/s | 6 m/s | 6 m/s | 11 m/s | 12 m/s | 11 m/s | 12 m/s |

Obtained DOFs were imported into CAD software. The coordinates of the CPs for further analysis were obtained as the centre of vectorized circles with a diameter of 20 cm, as shown in Figure 3.

Since the quality of the image itself not only depends on the number of pixels in an image, but also on other characteristics such as the assumption that the larger the sensor the better the quality and the dependence of quality on-screen aperture and shutter speed (screen aperture and shutter speed are in a reciprocal relationship), it can be assumed that the images taken with the senseFly eBee Plus, DJI Phantom 4 Pro V2.0 and Phantom 4 RTK sensors will be of a higher quality than the images taken with the DJI Phantom 4 sensor. Furthermore, as a consequence, the coordinates obtained from the DOF map obtained

from higher-quality shots will be more accurate and precise. However, the flight planning software is somewhat automated and does not allow full control of all camera settings; the shutter speed and aperture were set relative to the UAS speed, flight height, illumination and sensor sensitivity to keep the expected image motion blur within the sensor pixel size. Manual adjustments of aperture, shutter speed and ISO sensitivity would provide better insights into the impact of photo quality.

**Table 2.** UAS camera specifications.

|  | DJI Phantom 4 | DJI Phantom 4 Pro V2.0/DJI Phantom 4 RTK | senseFly eBee Plus |
|---|---|---|---|
| **Sensor** | CMOS | CMOS | CMOS |
| **Resolution** | 12.4 MP | 20.0 MP | 20.0 MP |
| **Sensor size** | ¹/₂.3″ | 1″ | 1″ |
| **Focal length** | 3.6 mm | 8.8 mm | 10.6 mm |
| **Field of view (FOV)** | 94° | 84° | 154° |
| **Aperture setting** | f/2.8 | f/2.8-f/11 | f/2.8-f/11 |
| **Shutter speed** | Electronic 8-1/8000 s | Electronic 8-1/8000 sMechanical 8-1/2000 s | Mechanical 1/500–1/2000 s |
| **GNSS RTK sensor** | + | −/+ | + |

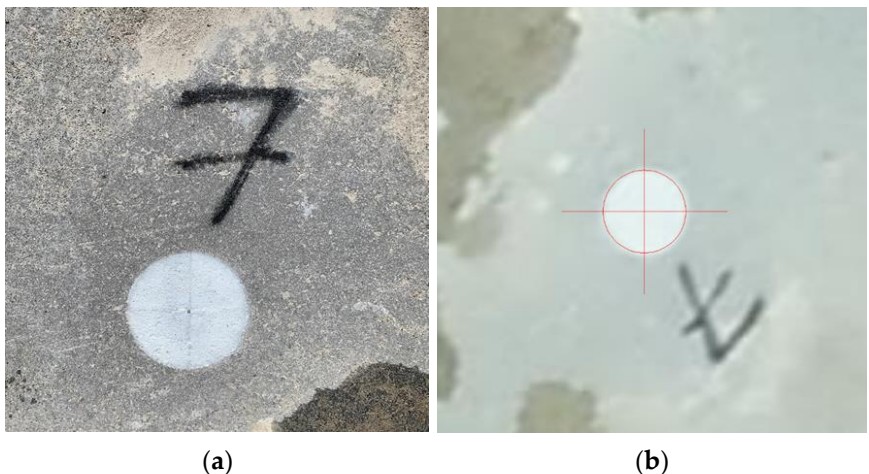

(**a**)                                    (**b**)

**Figure 3.** Checkpoint drawn in the field (**a**); CP vectorized on the DOF map (**b**).

To demonstrate how the use of UAS for geodetic surveying works in practice, a new site for conducting an experiment was established in the city of Jastrebarsko, Croatia (Figure 4). The study site consisted of 7 GCPs and covered 27 cadastral parcels with 19 properties having houses. The new study site was quite different from the first one because the conditions changed from a flat asphalt surface to an urban area. The State Geodetic Administration issued a decision approving aerial photogrammetry of the second study site. Four streets surrounded the area, and the GCPs were drawn in the streets using a previously made point model and painted with pink spray on the asphalt or concrete surface as circles with a diameter of 40 cm, with a survey marker in the middle. All GCPs were measured by the GNSS RTK method using the Trimble R12 receiver, which in RTK measurement mode has a horizontal accuracy of 8 mm + 0.5 ppm, and a vertical accuracy of 15 mm + 0.5 ppm [26]. GCPs were measured in two independent repetitions where each repetition had three consecutive measurements, and each measurement lasted 30 s (epoch). In accordance with the Ordinance on the method of performing basic geodetic work [24], all GCPs were measured again in the same way after two hours. Additionally, 48 points of the palisade fences in the field were measured by the GNSS RTK method.

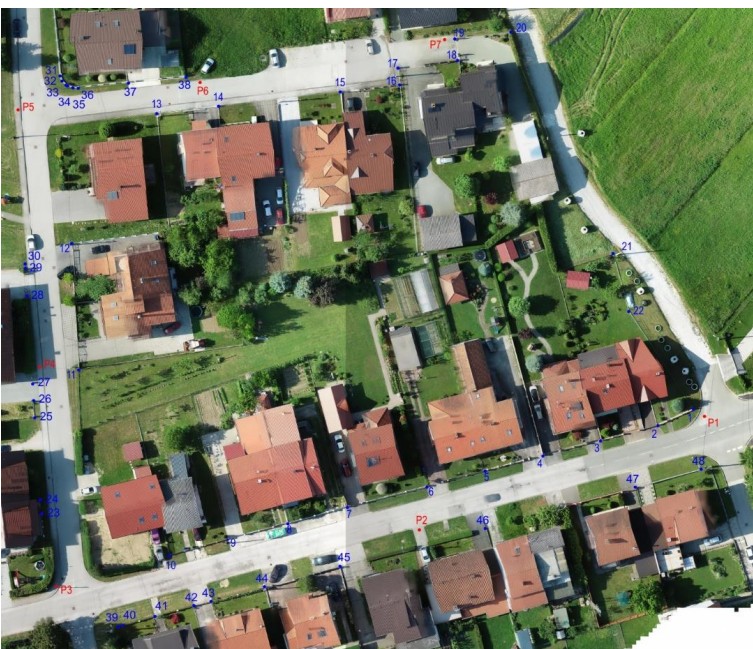

**Figure 4.** Study site in the city of Jastrebarsko, Croatia.

A UAS photogrammetry survey of the study area was performed using DJI Phantom 4 Pro V2.0 at a flight height of 60 m, using the same flight settings as in the first study site in terms of the overlap and camera angle. The results using Phantom 4 Pro V2.0 with GCPs at the same height were also shown in the work [27], where the correlation between flight height and obtained accuracy was also determined. Furthermore, the obtained accuracy satisfies the cadastral requirements of the Republic of Croatia regulations, i.e., the requirements set in Article 52 of the Ordinance on geodetic studies [17]. Therefore, Phantom 4 Pro V2.0 was used for the aerial photogrammetry survey at the second study site. The absolute orientation of the model was carried out using GCPs and a DOF map was constructed. Georeferencing of the model was carried out using GCPs because the results from the first study site showed that the absolute accuracy using GCPs is more accurate than using an RTK sensor. The coordinates of all 48 measured fence points were also obtained from the DOF map. In Figure 4, the fence points are marked with blue and the GCPs with red colour.

*Accuracy Analysis of the Results*

The Ground Sampling Distance (GSD) is a basic factor that defines the accuracy of aerial photography with a digital camera. This value represents the distance of the centres of two adjacent pixels observed on the ground. In general, if the recording was carried out correctly and all the sensors were calibrated, then it is assumed that the model is constructed correctly. In that case, relative accuracy between two points of such a constructed model (whether orthophoto or 3D model) is expected to be in the range of 1 to 3 times the GSD. The expected absolute accuracy of the model is lower by 1 to 2 times the GSD value in the horizontal E and N axis and by 1 to 3 times the GSD value in the vertical H axis [3]. According to research conducted by [28], expected relative and absolute accuracy are stated. As the accuracy of the coordinates in the projection plane was compared with the absolute coordinates in this study, accuracy below twice the GSD value was expected. The GSD value primarily depends on the camera specifications and flight altitude, and Table 2 shows the expected absolute accuracy for the four UAS used as a function of flight height and achievable GSD. The GSD values listed in Table 3 were obtained after model processing in the 3Dsurvey software.

**Table 3.** Expected absolute accuracy based on flight height and achievable GSD at different flight heights.

| Flight Height | DJI Phantom 4 | | DJI Phantom 4 Pro V2.0/DJI Phantom 4 RTK | | senseFly eBee Plus | |
|---|---|---|---|---|---|---|
| | GSD (cm) | Expected Absolute Accuracy (2xGSD) (cm) | GSD (cm) | Expected Absolute Accuracy (2xGSD) (cm) | GSD (cm) | Expected Absolute Accuracy (2xGSD) (cm) |
| 60 m | 2.6 | 5.2 | 1.6 | 3.2 | 1.7 | 3.4 |
| 120 m | 5.0 | 10.0 | 3.3 | 6.6 | 3.1 | 6.2 |

Accuracy analysis of aerial photogrammetry was performed by considering the survey data collected by classical surveying methods as true values and comparing them with the coordinates obtained by the aerial photogrammetry method from the DOF map to establish horizontal precision and accuracy of the results and the dependence of precision and accuracy regarding image sensor characteristics, flight heights and data obtained from UAS GNSS and IMU sensors. In order to obtain statistical indicators of coordinate errors along the coordinate axis, for each point the adjusted coordinates obtained by GNSS RTK measurement and the polar method were subtracted from the coordinates obtained based on the DOF map and based on these coordinate differences ($\varepsilon E$ and $\varepsilon N$) minimum, maximum, range, mean absolute error (MAE), root mean square error (RMSE) and mean square error (MSE) were calculated. The equations used for the calculation of MAE, RMSE and MSE are listed below, where n is the number of measurements.

$$MAE = \frac{1}{n} \sum_{1}^{n} |\varepsilon_i| \tag{1}$$

$$RMSE = \sqrt{\frac{1}{n} \sum_{1}^{n} \varepsilon_i^2} \tag{2}$$

$$MSE = \frac{1}{n} \sum_{1}^{n} \varepsilon_i^2 \tag{3}$$

Additionally, the elements of the 95% confidence ellipse were determined, and the confidence ellipses were drawn and marked in red.

Accuracy is defined as the degree of closeness of the measured data to the true values while precision is defined as the degree of mutual agreement of measurement data during repeated measurement of the same quantity. The difference between accuracy and precision of measurements is shown in Figure 5.

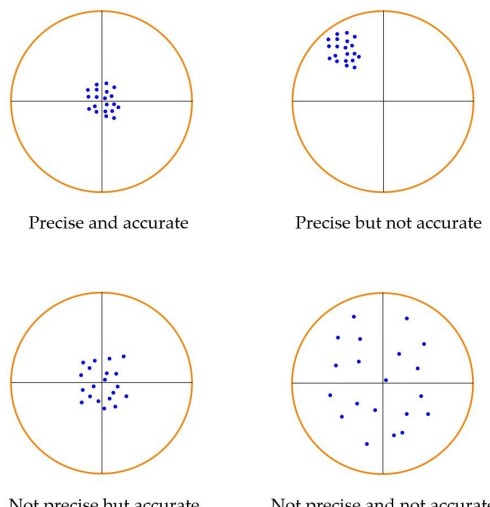

**Figure 5.** Precision versus accuracy.

## 3. Results

### 3.1. Testing Results from the First Study Site, Established Polygon, in Samobor, Croatia

The elements of the 95% confidence ellipse were determined, and the confidence ellipses were drawn and marked in red in Figures 6–10 for each project. According to Article 52 of the Ordinance on geodetic studies [17], "The quality of data of field survey of breakage points of parcel boundaries and other boundaries of cadastral parcels and buildings and other structures for the purposes of geodetic study, whose data are recorded in the cadastral register, is determined by the area of confidence for horizontal coordinates with 95% probability standard position accuracy up to 0.10 m". Regarding that article, a circle of 10 cm radius was visualized along the 95% confidence ellipses and marked in yellow in Figures 6–10. The reason for this was to present, i.e., to visualize, that the achieved quality of the obtained DOF map obtained by the UAS photogrammetric survey is suitable for cadastral survey purposes. Statistical indicators were calculated and are shown in Tables 4–8. In accordance with the Ordinance on the method of performing basic geodetic works [24], the elements of the standard error ellipses were multiplied by a factor of 2,45 to obtain parameters for the 95% confidence ellipse.

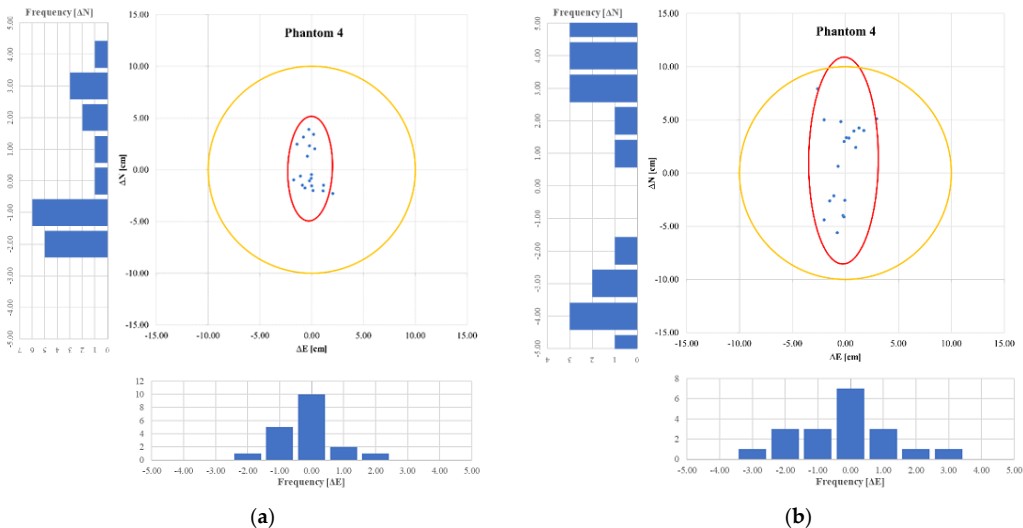

(**a**)                                        (**b**)

**Figure 6.** Confidence ellipse—Phantom 4 with GCPs at 60 m (**a**); Confidence ellipse—Phantom 4 with GCPs at 120 m (**b**).

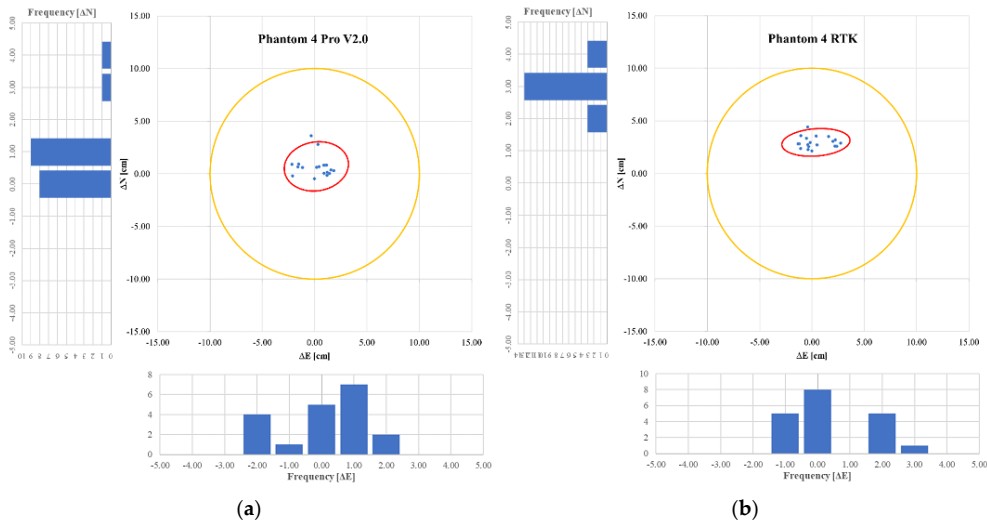

(**a**)                                        (**b**)

**Figure 7.** Confidence ellipse—Phantom 4 Pro V2.0 with GCPs at 60 m (**a**); Confidence ellipse—Phantom 4 RTK without GCPs (RTK) at 60 m (**b**).

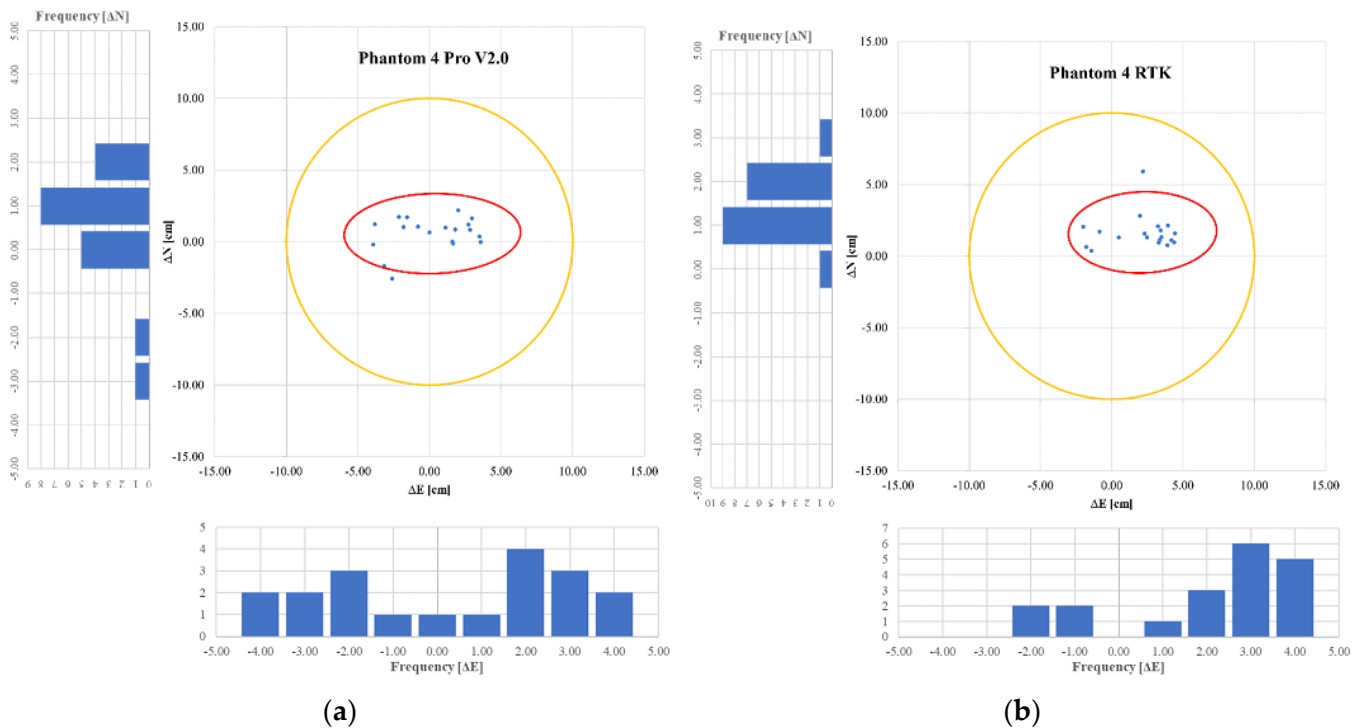

**Figure 8.** Confidence ellipse—Phantom 4 Pro V2.0 with GCPs at 120 m (**a**); Confidence ellipse—Phantom 4 RTK without GCPs (RTK) at 120 m (**b**).

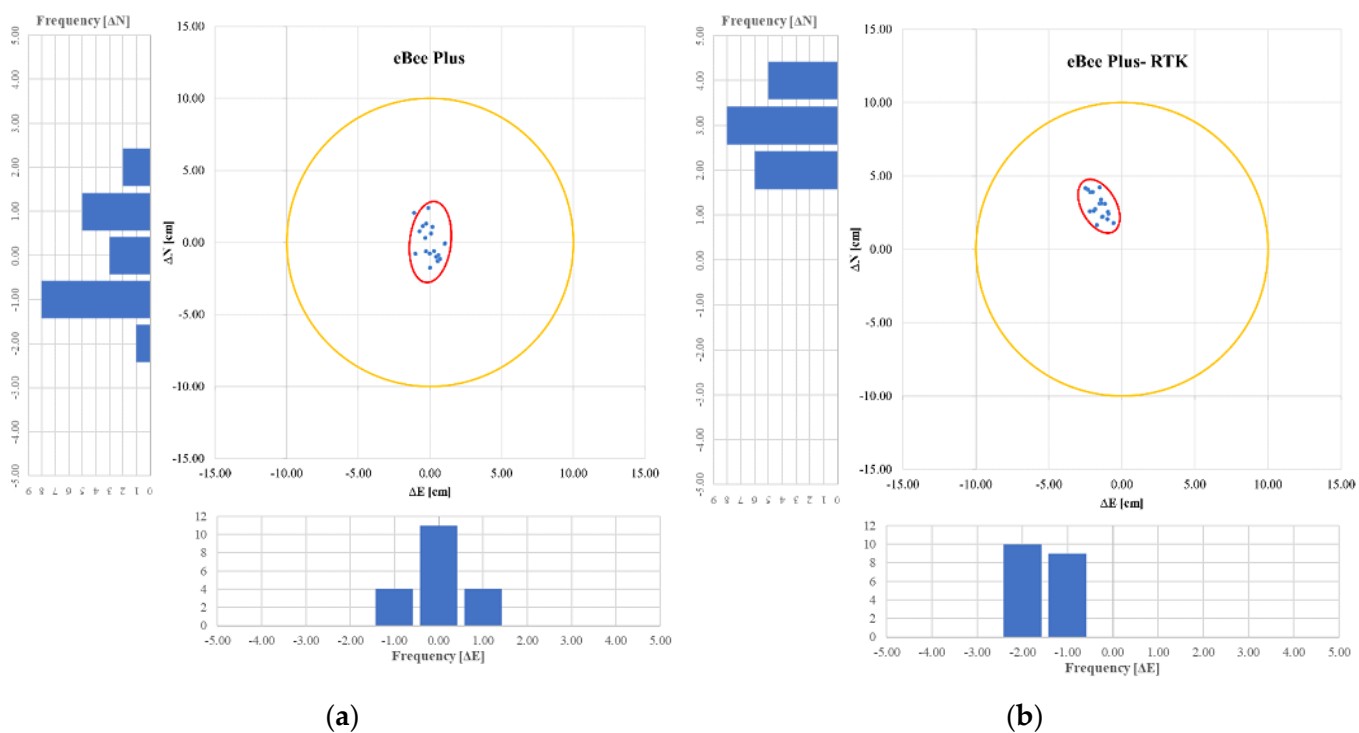

**Figure 9.** Confidence ellipse—eBee with GCPs at 60 m (**a**); Confidence ellipse—eBee without GCPs (RTK) at 60 m (**b**).

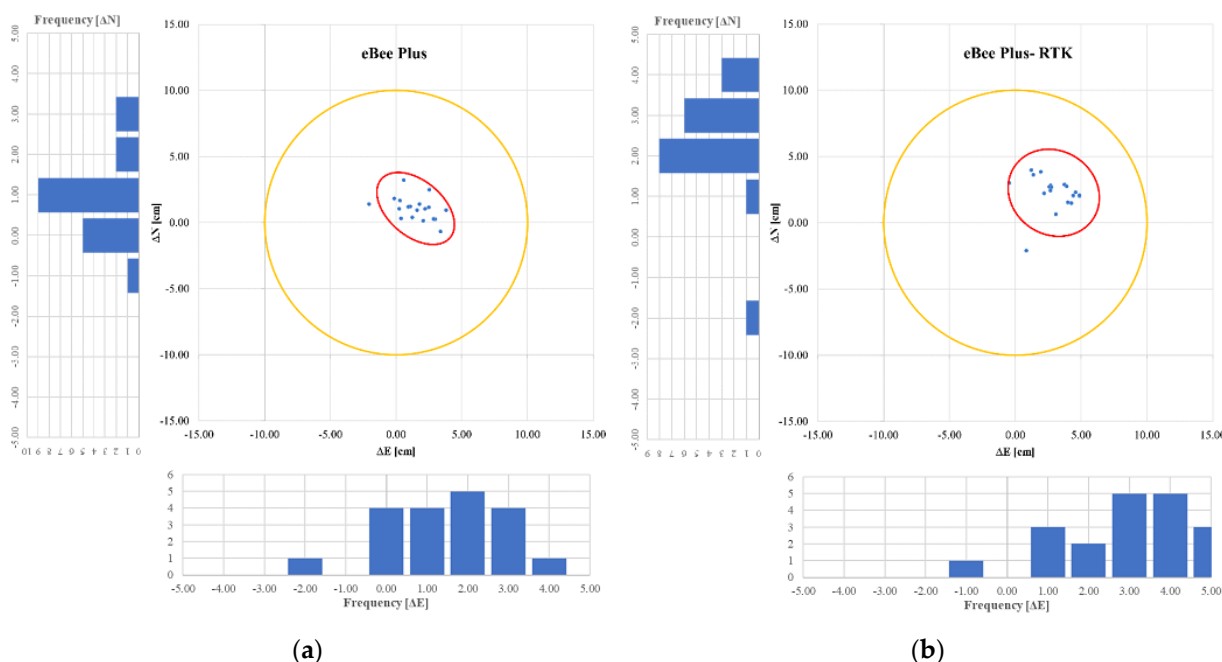

**Figure 10.** Confidence ellipse—eBee with GCPs at 120 m (**a**); Confidence ellipse—eBee without GCPs (RTK) at 120 m (**b**).

**Table 4.** UAS DJI Phantom 4—achieved data quality with the use of GCPs.

|  | 60 m | | 120 m | |
|---|---|---|---|---|
|  | εE [cm] | εN [cm] | εE [cm] | εN [cm] |
| Minimum | −1.7 | −2.3 | −2.6 | −5.6 |
| Maximum | 2.0 | 3.9 | 2.9 | 7.9 |
| Range | 3.8 | 6.2 | 5.6 | 13.5 |
| Mean absolute error (MAE) | 0.7 | 1.9 | 1.0 | 3.8 |
| Root mean square error (RMSE) | 0.9 | 2.1 | 1.4 | 4.1 |
| Mean square error (MSE) | 0.8 | 4.3 | 1.8 | 17.1 |
| **Elements of the 95% Confidence ellipse [cm]** | | | | |
| Semi-major axis A | 5.1 | | 9.7 | |
| Semi-minor axis B | 2.2 | | 3.3 | |
| Semi-major axis inclination angle Θ | 1.78° | | 0.44° | |

**Table 5.** UAS DJI Phantom 4 Pro V2.0—achieved data quality with the use of GCPs.

|  | 60 m | | 120 m | |
|---|---|---|---|---|
|  | εE [cm] | εN [cm] | εE [cm] | εN [cm] |
| Minimum | −2.2 | −0.5 | −3.9 | −2.6 |
| Maximum | 1.8 | 3.6 | 3.6 | 2.2 |
| Range | 4.0 | 4.1 | 7.5 | 4.8 |
| Mean absolute error (MAE) | 1.1 | 0.8 | 2.3 | 1.1 |
| Root mean square error (RMSE) | 1.3 | 1.2 | 2.5 | 1.3 |
| Mean square error (MSE) | 1.6 | 1.4 | 6.4 | 1.6 |
| **Elements of the 95% Confidence ellipse [cm]** | | | | |
| Semi-major axis A | 3.1 | | 6.2 | |
| Semi-minor axis B | 2.3 | | 2.8 | |
| Semi-major axis inclination angle Θ | 80.75° | | 88.76° | |

**Table 6.** UAS DJI Phantom 4 RTK—achieved data quality without the use of GCP (RTK).

| | 60 m | | 120 m | |
|---|---|---|---|---|
| | εE [cm] | εN [cm] | εE [cm] | εN [cm] |
| Minimum | −1.3 | 2.1 | −2.0 | 0.4 |
| Maximum | 2.7 | 4.4 | 4.4 | 5.9 |
| Range | 4.1 | 2.3 | 6.4 | 5.5 |
| Mean absolute error (MAE) | 1.1 | 3.0 | 2.8 | 1.7 |
| Root mean square error (RMSE) | 1.4 | 3.0 | 3.0 | 2.0 |
| Mean square error (MSE) | 1.9 | 9.0 | 9.2 | 4.1 |
| | **Elements of the 95% Confidence ellipse [cm]** | | | |
| Semi-major axis A | 3.3 | | 5.2 | |
| Semi-minor axis B | 1.3 | | 2.8 | |
| Semi-major axis inclination angle Θ | 85.83° | | 88.04° | |

**Table 7.** UAS senseFly eBee Plus—achieved data quality with the use of GCPs.

| | 60 m | | 120 m | |
|---|---|---|---|---|
| | εE [cm] | εN [cm] | εE [cm] | εN [cm] |
| Minimum | −1.1 | −1.8 | −2.1 | −0.7 |
| Maximum | 1.0 | 2.4 | 3.8 | 3.2 |
| Range | 2.1 | 4.2 | 5.9 | 3.9 |
| Mean absolute error (MAE) | 0.5 | 1.0 | 1.7 | 1.1 |
| Root mean square error (RMSE) | 0.6 | 1.1 | 2.0 | 1.4 |
| Mean square error (MSE) | 0.4 | 1.3 | 4.2 | 1.9 |
| | **Elements of the 95% Confidence ellipse [cm]** | | | |
| Semi-major axis A | 2.8 | | 3.4 | |
| Semi-minor axis B | 1.4 | | 2.1 | |
| Semi-major axis inclination angle Θ | 6.40° | | 129.88° | |

**Table 8.** UAS eBee—achieved data quality without the use of GCP (RTK).

| | 60 m | | 120 m | |
|---|---|---|---|---|
| | εE [cm] | εN [cm] | εE [cm] | εN [cm] |
| Minimum | −2.5 | 1.6 | −0.5 | −2.1 |
| Maximum | −0.6 | 4.2 | 4.9 | 4.0 |
| Range | 1.9 | 2.6 | 5.4 | 6.1 |
| Mean absolute error (MAE) | 1.6 | 2.9 | 3.0 | 2.5 |
| Root mean square error (RMSE) | 1.7 | 3.0 | 3.3 | 2.6 |
| Mean square error (MSE) | 2.7 | 9.3 | 10.6 | 6.8 |
| | **Elements of the 95% Confidence ellipse [cm]** | | | |
| Semi-major axis A | 2.0 | | 3.6 | |
| Semi-minor axis B | 1.2 | | 3.2 | |
| Semi-major axis inclination angle Θ | 150.10° | | 121.48° | |

In Figures 6–10 beside the 95% confidence ellipses and a circle of 10 cm radius according to Article 52 of the Ordinance on geodetic studies [17], the distribution of errors ΔE and ΔN by axis, i.e., their frequencies have been shown.

For each of the calculated DOFs, the difference in distance between points 7 and 19 and 11 and 15 were calculated using true values of point coordinates and values of the coordinates obtained from DOFs to define the scale of the calculated DOFs. Figure 11 shows

the selected distances. Point pair 7 and 19 was selected as the distance between the points stretched the most in the coordinate system axis E direction, while point pair 11 and 15 was selected as the distance between the points stretched the most in the coordinate system axis N direction. Those two distances were chosen as they correspond to the directions of the UAS's flight operations, if there are any inconsistencies in the model's scale, it is anticipated to be manifested after inspecting the two linear scales that are perpendicular to each other.

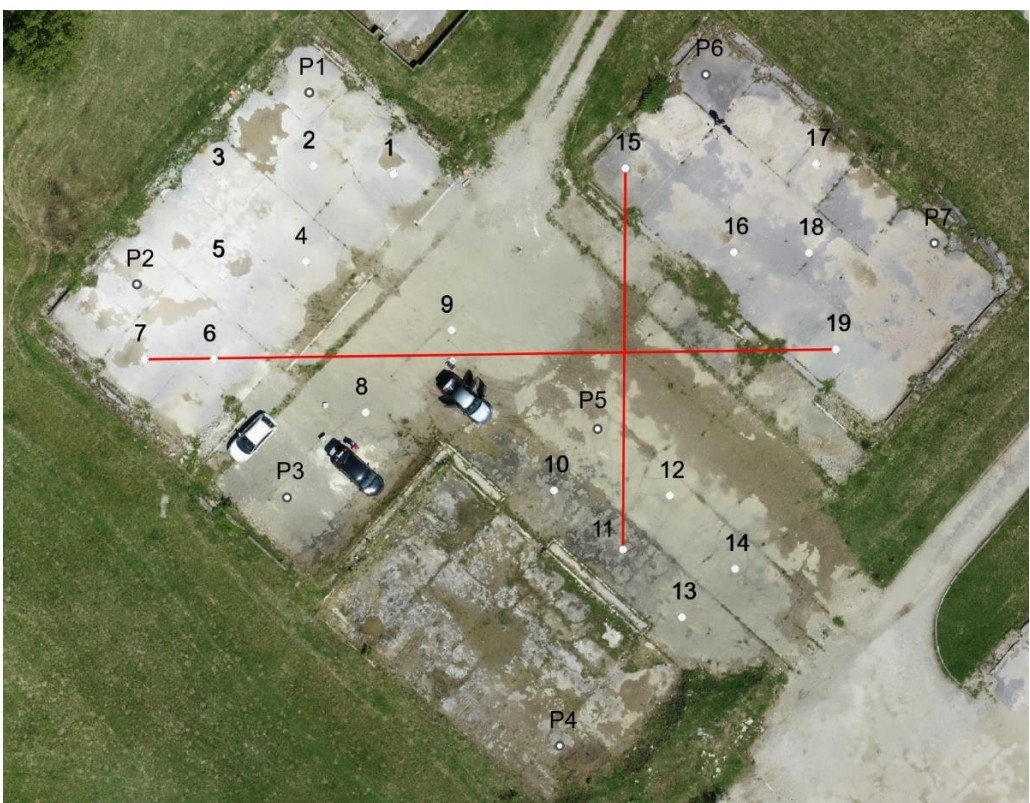

**Figure 11.** Selected distances between points.

The small values of the calculated distance differences from Table 9 indicate that DOFs have a very high relative accuracy between the points. The most extreme case of the differences between the two linear scales (which occurs in Phantom 4's model at the flight height of 120 m) reveals the linear scale in the direction E-W of 1:0.999 and the linear scale in the direction of N-S of 1:1.002. Even for the extremity, the difference between the linear scales is non-significant. Therefore, we can conclude that the scale(s) over the constructed DOFs are uniform and are 1.

**Table 9.** Calculated distance differences between point pairs 7 and 19, and 11 and 15.

| Flight Height | DJI Phantom 4 | | DJI Phantom 4 Pro V2.0 | | DJI Phantom 4 RTK | | senseFly eBee Plus (GCP/RTK) | |
|---|---|---|---|---|---|---|---|---|
| | Difference CP (7–19) (cm) | Difference CP (11–15) (cm) | Difference CP (7–19) (cm) | Difference CP (11–15) (cm) | Difference CP (7–19) (cm) | Difference CP (11–15) (cm) | Difference CP (7–19) (cm) | Difference CP (11–15) (cm) |
| 60 m | −2.0 | 4.9 | −1.0 | 0.1 | −0.6 | 0.4 | −0.5/−0.6 | 1.0/0.8 |
| 120 m | −3.9 | 7.3 | 3.1 | −0.5 | −4.0 | −1.1 | 0.6/3.8 | −0.4/−0.6 |

Map scales shown in Table 10 were calculated as the ratio of the distance between points calculated using true values of point coordinates and the values of the coordinates obtained from DOFs. All of the map scales were determined as 1:1.

**Table 10.** Calculated map scales.

| Flight Height | DJI Phantom 4 | | DJI Phantom 4 Pro V2.0 | | DJI Phantom 4 RTK | | senseFly eBee Plus (GCP/RTK) | |
|---|---|---|---|---|---|---|---|---|
| | Scale CP (7–19) | Scale CP (11–15) | Difference CP (7–19) (cm) | Difference CP (11–15) (cm) | Difference CP (7–19) (cm) | Difference CP (11–15) (cm) | Difference CP (7–19) (cm) | Difference CP (11–15) (cm) |
| 60 m | 1:1 | 1:1 | 1:1 | 1:1 | 1:1 | 1:1 | 1:1/1:1 | 1:1/1:1 |
| 120 m | 1:1 | 1:1 | 1:1 | 1:1 | 1:1 | 1:1 | 1:1/1:1 | 1:1/1:1 |

From the obtained results presented above, it can be concluded that the most accurate and precise results were obtained using eBee and Phantom 4 Pro V2.0 using GCPs at lower flight height, i.e., at 60 m. Additionally, from the plotted coordinate differences shown in Figures 6–10, it is notable that the accuracy of all measurements is very high but the results using GCPs are slightly more accurate. The same coordinate differences show the very high precision of the results in all cases except the results regarding UAS Phantom 4 at 120 m flight height. [17,27]

*3.2. Testing Results from the Second Study Site, a Small Settlement, in Jastrebarsko, Croatia*

Refs. [24,26] An accuracy analysis of the obtained DOF map performed by UAS aerial photogrammetry of the new study area was carried out by considering the survey data collected by the GNSS RTK method as true values and comparing them with the coordinates obtained by the aerial photogrammetry method from the DOF map to obtain horizontal precision and accuracy of the results. Statistical indicators were calculated and are shown in Table 11 and the elements of the 95% confidence ellipse were determined, and the confidence ellipse was drawn and marked in red in Figure 12.

**Table 11.** UAS DJI Phantom 4 Pro V2.0—data quality with the use of GCPs.

| | 60 m | |
|---|---|---|
| | $\varepsilon$E [cm] | $\varepsilon$N [cm] |
| Minimum | −6.5 | −5.7 |
| Maximum | 6.0 | 4.8 |
| Range | 12.5 | 10.5 |
| Mean absolute error (MAE) | 2.3 | 1.8 |
| Root mean square error (RMSE) | 2.9 | 2.2 |
| Mean square error (MSE) | 8.2 | 5.0 |
| | **Elements of the 95% Confidence ellipse [cm]** | |
| Semi-major axis A | 6.9 | |
| Semi-minor axis B | 5.4 | |
| Semi-major axis inclination angle $\Theta$ | 87.95° | |

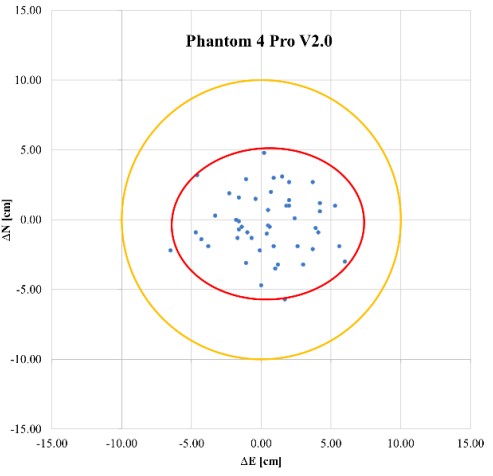

**Figure 12.** Confidence ellipse—Phantom 4 Pro V2.0 with GCPs at 60 m.

The shape of the plotted confidence ellipse in Figure 12 shows that the cadastral requirements of the Republic of Croatia regulations, i.e., the requirements set in Article 52 of the Ordinance on geodetic studies [17], were satisfied using the previously described setup. To sum up, very accurate and precise results were obtained even though the survey conditions were not as "laboratory" as in the first field test.

## 4. Discussion

The first step of this study was focused on determining an achievable accuracy depending on the absolute orientation of the models and used cameras based on the comparison of the RMSE values at the CP. For all performed projects, the final 2D accuracies were below two times the GSD, and the lowest achievable accuracy was 4.1 cm, as shown in Table 4 (DJI Phantom 4 model). In all other projects, better results, i.e., higher accuracies, were reached.

From the values of the accuracy assessment measures listed in Tables 4–8, it can be noticed that a higher measurement accuracy was obtained at lower flight heights. Additionally, from the same measurements, it is obvious that at the same flight altitudes the higher measurement accuracy is obtained from the DOF map generated from the images with a higher quality sensor. Elements and shapes of calculated and plotted confidence ellipses indicate higher measurement precision at lower flight heights and from the DOF map generated from the images with a higher quality sensor. Additionally, shapes of calculated and plotted confidence ellipses (Figures 6–10) indicate that the same measurement precision, but not accuracy, is obtained using different georeferencing methods, i.e., absolute orientation of the models at the same flight heights.

Regarding the results from the second field test in the city of Jastrebarsko, the results are not at the same level as the results obtained at the same flight height and with the same UAS from the first study area in the city of Samobor, but are still quite accurate regarding the fact that most of the investigated fences have post caps on them, which made it harder to decipher the corners of the fences on DOF map, as well as to measure them with the GNSS RTK method (Figure 13).

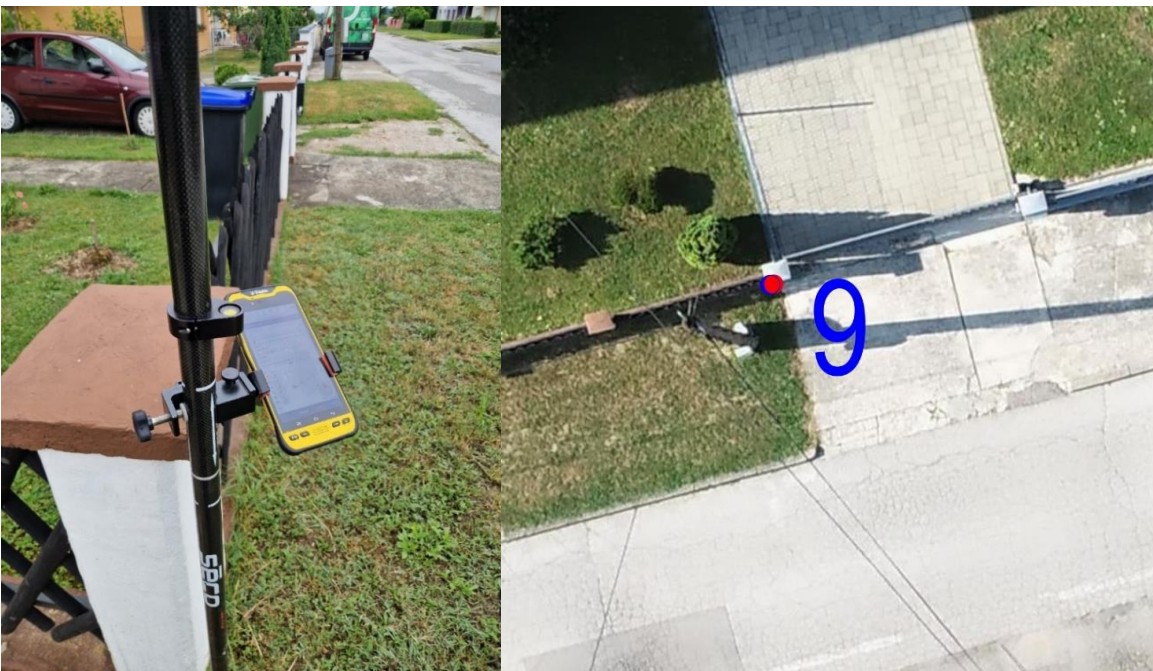

**Figure 13.** Concrete fence with a post cap in the field (**left**); Vectorized fence point shown on the DOF map with red and GNSS RTK measured point with blue colour (**right**).

As suggested in the work by [3], the GCPs in both study sites were distributed evenly around the whole study site. Additionally, GCPs were distributed over the edges of the study sites as discussed in [4,5] and large image overlap has been used as suggested in [7]. The second study site showed that UASs can be efficiently used for accurate cadastral survey purposes. The study showed that the cadastral requirements of the Republic of Croatia were met, as well as in the work regarding the cadastral requirements of the Czech Republic [8], Poland [9] and Switzerland [10]. Furthermore, this study showed that it is possible to design a methodology that fits FIG's "fit-for-purpose" approach in terms of the use of UASs for cadastral survey purposes in the countries where regulations require more accurate results in contrast with the approach designed for developing countries by [11].

In this study, the models were reconstructed using solely RTK solution or georeferenced using densely distributed high-quality GCPs. In such a manner, the quality of the generated models would display the main differences between the two approaches. From the presented numerical results and graphs, it is apparent that the RTK solution is less accurate. To augment the absolute accuracy of the models using only the RTK solution, the process of implementing only 1 GCP would be beneficial. This approach would lead to a significant increase in vertical accuracy, as well as an increase in absolute 2D accuracy by translating the E and N axis in the horizontal system. This part was omitted from the study as the main intention was to present the results and distinction between the two uniquely disparate approaches.

## 5. Conclusions

Performing an aerial photogrammetry survey with UAS using a relatively dense structure of GCPs established in the study site will result in reliable and survey-grade accuracy. This accuracy satisfies the cadastral requirements of the Republic of Croatia regulations. As expected, there is a high correlation between the quality of digital image sensors and the accuracy of the model, as well as a high negative correlation between height flights and the accuracy; the standard deviation of horizontal coordinates tends to increase with flight altitude. It should also be noted that the first study site in this research was set in almost "laboratory conditions", with the field being a flat asphalt surface. The second study site had discrepancies from the "laboratory conditions" in terms of fence caps where it was harder to decipher the real positions of the fence corners from the DOF map. Nevertheless, results from the second field test also indicate that high measurement accuracy can be achieved using a DOF map obtained from UAS aerial photogrammetry for cadastral survey purposes. The findings support the development of various UAS-based cadastral survey workflows to deliver efficient and reliable datasets using optimal flight height and image sensors to meet land administration system "fit-for-purpose" requirements. There is still work to be done to test UAS in different conditions such as dense urban areas. Additionally, only horizontal coordinates were analysed in this study, so, still, the vertical accuracy obtained from UAS aerial photogrammetry needs to be investigated.

**Author Contributions:** Conceptualization, H.S., R.P.; methodology, R.P.; software, H.S. and F.R.; validation, H.S., R.P., H.T. and F.R.; formal analysis, H.S., R.P. and F.R.; investigation, H.S., R.P. and F.R.; resources, H.S. and R.P.; writing—original draft preparation, H.S. and R.P.; writing—review and editing, R.P., H.T. and F.R.; visualization, H.S. and F.R.; supervision, R.P. and H.T.; project administration, R.P. All authors have read and agreed to the published version of the manuscript.

**Funding:** This research received no external funding.

**Institutional Review Board Statement:** Not applicable.

**Informed Consent Statement:** Not applicable.

**Data Availability Statement:** The data presented in this study are available on request from the corresponding author.

**Conflicts of Interest:** The authors declare no conflict of interest.

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
