# Peer review of "Influence of Flight Height and Image Sensor on the Quality of the UAS Orthophotos for Cadastral Survey Purposes"

_land, doi:10.3390/land11081250_

Round 1
Reviewer 1 Report
The manuscript focused on the quality of orthophotos for cadastral survey and the related influence of flight height. The manuscript is well written but there are a lack in some parts. If authors well follow the advice I will recomend this article for pubblication on Land.
Firstly, I suggest consider this work in the introduction:
- https://doi.org/10.5194/isprsarchives-XXXVIII-1-C22-57-2011
- https://doi.org/10.3390/rs12213542
- https://doi.org/10.1007/s12145-017-0314-6
- https://doi.org/10.3390/rs12213625
In material and methods sections I suggest you to introduce
- the flight parameters (height, speed, etc)
- the software adopted to precessing the data and describe in detail each settings.
Finally describe whihc tool you adopted to compute RMSE in conclusion section.
Author Response
Response to Reviewer 1 Comments
Reviewer 1
Dear reviewer,
thank you for your review. We very much appreciate your valuable comments and detected shortcomings in our article which are now solved in this revised version. The corrections made according to your comments and suggestions improved our article and are highlighted with red colour in revised version of the article.
The manuscript focused on the quality of orthophotos for cadastral survey and the related influence of flight height. The manuscript is well written but there are a lack in some parts. If authors well follow the advice I will recomend this article for pubblication on Land.
Firstly, I suggest consider this work in the introduction:
- https://doi.org/10.5194/isprsarchives-XXXVIII-1-C22-57-2011
- https://doi.org/10.3390/rs12213542
- https://doi.org/10.1007/s12145-017-0314-6
- https://doi.org/10.3390/rs12213625
Thank you for the recommended articles! Relevant literature is added in the introduction, while one was already cited in our work.
In material and methods sections I suggest you to introduce
- the flight parameters (height, speed, etc)
- the software adopted to precessing the data and describe in detail each settings.
Finally describe whihc tool you adopted to compute RMSE in conclusion section.
Thank you for these suggestions! All suggestions were accepted and implemented in the article.
Thank you for your valuable comments and detected shortcomings which improved our article!
Yours sincerely,
Authors.

Reviewer 2 Report
Dear authors,
Thanks for the manuscript. The study is well designed and the article is mostly easy to read but additional work needs to be done before the study is suitable to publish (in my opinion). Some parts of the article need to be clarified and restructured (some method parts presented in the results section) and with the data collected it is possible to conduct some further analyses to strengthen the results.
Introduction:
The aim of the study is not clearly presented. Add aim (explicitly) and what research questions the study is answering last in the introduction.
Methods and study areas:
Study area. Only the study area Samobor in Croatia is presented in the methods section. Both study areas must be presented in the methods section. Now the second study area (Jastrebarsko) pop up as a surprise in the results which is confusing to a reader. Clearly state that two study areas were used and explain the differences. Why did you only fly with one UAV in Jastrebarsko and how did you decide what UAV to use there? Now it gives the impression you did two studies and merged them to one manuscript.
p. 7, l. 237 – 258. The first section of the results is more related to methods. Move it to methods and add a detailed description of both study areas (as mentioned above).
p. 7, l. 259 – p. 8, l. 270. This describes the statistics used to evaluate the accuracy. It belongs to the methods. Add a section about accuracy assessment/evaluation in the methods section where you clearly described how the results were evaluated. Also clarify precision versus accuracy in that section.
Results:
Confidence ellipses are very nice illustrations but they do not provide any information about the spatial distribution of the deviations. Are the deviations systematic, e.g. are control points with positive εE grouped or randomly distributed? You need to add a section about the spatial distribution of the deviations.
Related to the spatial distribution of deviation is the scale of the DOF. Is the scale constant in the DOF? You have measured CPs with high accuracy, you can easily select a set of CPs and evaluate the distances between CPs in the DOFs to get an indication if the scale is constant. Add this analysis to the results.
Include a section about precision and accuracy. It will be interesting for the comparison between RTK UAVs and GCP georeferenced DOF. It could be shortly mentioned in the results and then elaborated on in the discussion.
Discussion:
The discussion is short and the study is not placed in context with other studies. You have no comparisons with the studies presented in the background. How do your results agree with previous studies? What additional knowledge do you contribute with? Elaborate on the precision – accuracy discussion. Would one single GCP help to improve accuracy when RTK UAV only was used? Did you try to improve the absolute accuracy of the DOFs created from the RTK enables UAVs by a single GCP? Extend the discussion.
Minor comments:
p. 6, l. 215. Have a more informative table caption.
p. 7, l. 248 – 249. This sentence is confusing. Clarify.
Author Response
Response to Reviewer 2 Comments
Reviewer 2
Dear reviewer,
thank you for your review. We very much appreciate your valuable comments and detected shortcomings which are now solved in this revised version. The corrections made according to your comments and suggestions improved our article and are highlighted with red colour.
Dear authors,
Thanks for the manuscript. The study is well designed and the article is mostly easy to read but additional work needs to be done before the study is suitable to publish (in my opinion). Some parts of the article need to be clarified and restructured (some method parts presented in the results section) and with the data collected it is possible to conduct some further analyses to strengthen the results.
Thank you for this comment. We are aware that the article needs some improvements so your suggestions, in particular regarding the clarification and restruction of some parts will strengthen our article. We hope that with this revised version we will meet your expectations.
Introduction:
The aim of the study is not clearly presented. Add aim (explicitly) and what research questions the study is answering last in the introduction.
Thank you! We have added explicitly the aim of our study in the introduction.
Methods and study areas:
Study area. Only the study area Samobor in Croatia is presented in the methods section. Both study areas must be presented in the methods section. Now the second study area (Jastrebarsko) pop up as a surprise in the results which is confusing to a reader. Clearly state that two study areas were used and explain the differences. Why did you only fly with one UAV in Jastrebarsko and how did you decide what UAV to use there? Now it gives the impression you did two studies and merged them to one manuscript.
Yes, you are absolutely right regarding these facts about the two study areas. Somehow, we didn’t manage to connect them clearly. What we wanted to do and we did that is the following. The first study area in Samobor was our testing polygon, where we used several UAVs with different image sensors characteristics at different flight heights over the same polygon using GCP or UAV RTK sensor. With conducted analysis from this study area we made conclusions and after that we wanted to perform another test over small city area in Jastrebarsko to further confirm our conclusions from the first study area. So, based on the results from the first study area we concluded what is the best UAV, i.e., the best image sensor at what flight height using GCP or UAV RTK sensor for conducting the best orthophoto for the purpose of cadastral survey. In this revised version we described this in detail and moved the second study area Jastrebarsko to methods section. Thank you for highlighting this shortcomings!
- 7, l. 237 – 258. The first section of the results is more related to methods. Move it to methods and add a detailed description of both study areas (as mentioned above).
- 7, l. 259 – p. 8, l. 270. This describes the statistics used to evaluate the accuracy. It belongs to the methods. Add a section about accuracy assessment/evaluation in the methods section where you clearly described how the results were evaluated. Also clarify precision versus accuracy in that section.
You are right. The layout you suggested is way more understandable. We have adopted all your suggestions in our work. We have also added one nice figure which to our opinion clearly differs and clarifies the difference between the accuracy and precision. We hope you will like this. Thank you.
Results:
Confidence ellipses are very nice illustrations but they do not provide any information about the spatial distribution of the deviations. Are the deviations systematic, e.g. are control points with positive εE grouped or randomly distributed? You need to add a section about the spatial distribution of the deviations.
Related to the spatial distribution of deviation is the scale of the DOF. Is the scale constant in the DOF? You have measured CPs with high accuracy, you can easily select a set of CPs and evaluate the distances between CPs in the DOFs to get an indication if the scale is constant. Add this analysis to the results.
Include a section about precision and accuracy. It will be interesting for the comparison between RTK UAVs and GCP georeferenced DOF. It could be shortly mentioned in the results and then elaborated on in the discussion.
Thank you. All your comments and suggestions regarding the results section have been adopted in our work. These improvements have really strengthened the results and the article in general.
Discussion:
The discussion is short and the study is not placed in context with other studies. You have no comparisons with the studies presented in the background. How do your results agree with previous studies? What additional knowledge do you contribute with? Elaborate on the precision – accuracy discussion. Would one single GCP help to improve accuracy when RTK UAV only was used? Did you try to improve the absolute accuracy of the DOFs created from the RTK enables UAVs by a single GCP? Extend the discussion.
Thank you. We see it now also, so we have extended the discussion based on your comments and suggestions. The only thing we haven’t implemented is the use of a single GCP because this approach wasn’t one of the topics of our study. The use of one single GCP would mainly improve the vertical component of the accuracy and also the horizontal increase in absolute 2D accuracy by translating the E and N axis in the horizontal system but without rotation and scale of the system. This approach was omitted from our study as the main intention was to present the results and distinction between the two uniquely different approaches. Nevertheless, we did comment this approach in the discussion, and it could also be our potential future work.
Minor comments:
- 6, l. 215. Have a more informative table caption.
- 7, l. 248 – 249. This sentence is confusing. Clarify.
We have made minor improvements regarding this comments.
Thank you for your valuable comments and detected shortcomings which improved our article!
Yours sincerely,
Authors.

Round 2
Reviewer 1 Report
The authors well adress the suggestion given therefore the manuscript can go ahead in the pubblication phase.
Reviewer 2 Report
Dear Authors,
Thanks for the revised version of the manuscript. I think it is good and it will be interesting to readers.